# Position: ICML Should Treat Hosted LLM APIs as Versioned Dependencies and Require Drift-Audit Artifacts

**Utsav Gupta** [1]

## Abstract

This position paper argues that ICML should require a minimal drift-audit artifact for papers whose main claims materially rely on hosted LLM APIs. Hosted APIs can change behavior over time, undermining the scientific interpretability of results even when evaluation code and prompts are held fixed. While existing proposals address API contracts and change reporting, there is not yet a widely adopted, venue-aligned standard for attaching a minimal drift-audit artifact to results that rely on hosted endpoints. The paper proposes a lightweight artifact consisting of a small suite of invariant-checking probes (e.g., schema, tool-call, or refusal invariants), machine-readable provenance metadata, and a rerun script that can detect and characterize post-publication behavioral drift at bounded cost. It further argues that provider-side behavioral versioning and machine-readable changelogs are enabling infrastructure that would make drift-aware reporting more reliable and less burdensome. The paper concludes with concrete actions for conferences, providers, and tool builders, and with falsifiable predictions about improved replication stability and reduced time-to-diagnosis when results stop reproducing.

## 1. Introduction

This paper takes a position on a growing reproducibility failure mode in ML research: dependence on black-box, hosted LLM APIs that drift over time.

**Position:** *ICML should require a minimal drift-audit artifact for papers whose main claims materially rely on hosted LLM APIs.* Provider-side behavioral versioning and changelogs are enabling infrastructure that supports this, but the venue artifact is the actionable core of this proposal.

The core issue is simple. In traditional ML experimentation, we can usually pin code versions, datasets, and (if needed) model checkpoints. But when the "model" is an externally hosted service, the true dependency is not a static artifact; it is a moving system. That creates a mismatch between our norms for reproducibility and the reality of how modern LLM-based results are produced.

This problem is compounded by the increasing importance of hosted LLM APIs across the research landscape. Papers in NLP, software engineering, AI safety, social science, and other fields now routinely depend on commercial API endpoints as core experimental infrastructure. The ML systems community has long recognized that external dependencies create "hidden technical debt" (Sculley et al., 2015), and that deploying ML in production raises distinctive challenges around monitoring, observability, and maintenance (Paleyes et al., 2022). Hosted LLM APIs bring these challenges directly into the research workflow: when the "production system" is also the research instrument, and when it changes without notice, the scientific record becomes fragile.

## 2. Evidence: Hosted APIs Drift, Deprecate, and Break Downstream Results

### 2.1. Behavior drift has been empirically observed

Chen et al. (2023) evaluate GPT-3.5 and GPT-4 service versions across time (March vs. June 2023) and report large changes in performance and behavior across tasks, concluding that continuous monitoring is needed. The key takeaway is that meaningful drift occurs even while the dependency is nominally "the same service."

### 2.2. Drift is a software engineering problem, not just a benchmarking curiosity

Ma et al. (2023) describe how silent updates and scheduled deprecations of LLM APIs can cause performance regressions and prompt brittleness, and argue that regression testing for LLMs requires fundamental changes compared to traditional testing. This aligns with what many researchers experience informally: prompt-dependent pipelines can be

[1]Stanford University, Stanford, CA, USA. Correspondence to: Utsav Gupta <vastu@stanford.edu>.

*Proceedings of the 43rd International Conference on Machine Learning*, Seoul, South Korea. PMLR 306, 2026. Copyright 2026 by the author(s).

surprisingly sensitive to small upstream shifts.

### 2.3. This phenomenon predates LLMs: MLaaS APIs shift too

Chen et al. (2021) show that commercial ML prediction APIs can shift over time and propose MASA, an adaptive sampling method to estimate confusion-matrix shifts more sample-efficiently than naive sampling. The broader lesson is that model shift in hosted APIs is a recurring phenomenon, and monitoring can be made cost-effective.

### 2.4. Deprecations are routine, and availability is not guaranteed

Providers explicitly document model and endpoint retirement. OpenAI documents model/endpoint deprecations with shutdown dates and replacements (OpenAI, 2024b), and maintains an API changelog (OpenAI, 2024a). Anthropic documents model deprecations and advises customers to update applications as older models retire (Anthropic, 2024). Google's Gemini API publishes release notes and a deprecations schedule with shutdown-date guidance (Google, 2024a;b).

For scientific reproducibility, this means the "same" endpoint name is often not a stable identifier, and the underlying served behavior may change or disappear entirely (OpenAI, 2024a;b; Anthropic, 2024; Google, 2024a;b).

### 2.5. Behavioral drift is an instance of a well-studied phenomenon

The problem this paper describes is not new in kind. *Concept drift*—the phenomenon where the statistical properties of a target variable change over time—has been studied extensively in the machine learning literature (Lu et al., 2020). What is new is the *setting*: when the drifting component is a hosted inference API that the researcher treats as a fixed function, and when the drift is caused not by a changing data distribution but by unannounced updates to the model, its decoding stack, or its safety filters. This reframes API behavioral drift as a specific, operationally distinct instance of concept drift: one where the "concept" is the provider's serving behavior, the "drift" is a deployment decision, and the downstream "learner" is a research pipeline that assumed stationarity.

### 2.6. Reproducibility failures span disciplines

The evidence is not confined to ML benchmarking. Barrie et al. (2025) examine replication for LLM-based research in political science and document that closed-model non-determinism, silent updates, and the inability to pin model versions undermine reproducibility; they recommend that researchers use open-weight, versionable models where

possible and report detailed provenance when they cannot. Thomas et al. (2026) measure the reliability of generative AI outputs in academic business research and find "jagged competencies"—inconsistent performance across tasks and over time—that complicate any claim of stable results. Kapoor & Narayanan (2023) argue that OpenAI's policies actively hinder reproducible research: rate limits, usage restrictions, and the absence of persistent model snapshots make it difficult to re-run experiments even shortly after publication.

These findings from political science, business research, and science policy reinforce the ML-internal evidence: **hosted API drift is a cross-disciplinary reproducibility problem**, not a niche benchmarking concern.

## 3. Why Existing Documentation Norms Are Not Enough

The community has excellent frameworks for documenting *static* artifacts:

- Model Cards for models (Mitchell et al., 2019) and Datasheets for datasets (Gebru et al., 2018) improved transparency and responsible use.

- Reward Reports propose "living" documentation to track deployed and iteratively updated learning systems (Gilbert et al., 2023).

- Holistic evaluation frameworks such as HELM (Liang et al., 2022) and behavioral testing suites such as CheckList (Ribeiro et al., 2020) provide methods for systematically characterizing model capabilities at a point in time.

These are important, but hosted LLM APIs introduce a specific gap: **even perfect documentation at time $t$ does not guarantee interpretability at time $t + \Delta$** if the dependency drifts silently. We need a norm that explicitly treats "hosted model behavior over time" as part of the scientific object.

**The transparency gap is especially wide for closed models.** Liao & Vaughan (2023) present a human-centered research roadmap for AI transparency in the age of LLMs, noting that closed-model deployment creates fundamental barriers to the kind of transparency that scientific reproducibility requires. Liesenfeld et al. (2023) systematically track the openness, transparency, and accountability of instruction-tuned text generators and find that most commercially deployed models score poorly on all three dimensions. The NIST AI Risk Management Framework identifies transparency and documentation as core governance functions for AI systems, but does not prescribe venue-level reproducibility artifacts (NIST, 2023).

**Closed models compound reproducibility problems beyond drift.** Balloccu et al. (2024) document data contamination and evaluation malpractices in closed-source LLMs, showing that even benchmark results may be unreliable when training data is opaque. This means that researchers relying on closed APIs face two interacting threats: behavioral drift *and* evaluation validity concerns. A drift-audit artifact cannot solve data contamination, but it provides a concrete, checkable handle on one of the two failure modes.

**API drift is a form of hidden technical debt.** Sculley et al. (2015) identify "changing anything changes everything" as a core source of hidden technical debt in ML systems, and specifically flag external dependencies—data feeds, upstream models, and calibration layers—as high-risk components that can silently invalidate downstream assumptions. Hosted LLM APIs are a modern instance of exactly this pattern: they are external dependencies whose behavior can change without notice, accumulating technical debt in every downstream research pipeline that depends on them (Paleyes et al., 2022).

## 4. Related Work on Change Disclosure, Contracts, and Auditing

Several lines of work point toward practical solutions, but stop short of an end-to-end, venue-adoptable standard.

- **Change disclosure for ML updates.** Model Change-Lists propose methods and interfaces to characterize how ML models change across updates and how such updates can adversely affect users' systems (Eyuboglu et al., 2024).

- **Contracts for LLM APIs.** Romel et al. (2025) analyze hundreds of real-world contract violations, highlight LLM-specific contract types (format, policy constraints, streaming assembly, etc.), and recommend documentation improvements such as machine-readable schemas and a contract compatibility matrix.

- **Black-box auditing and monitoring.** Model Equality Testing frames monitoring as two-sample testing for distribution shifts (Gao et al., 2025). Logprob tracking enables cheaper continuous monitoring (Chauvin et al., 2025). Cai et al. (2025) show that software-only detection is unreliable against adversarial substitution—but this proposal targets cooperative reproducibility, not adversarial detection.

- **LLM auditing frameworks.** Mökander et al. (2024) propose three-layered LLM auditing (governance, model, application). The LDAA is an application-layer audit scoped to a paper's specific behavioral contract.

- **Ultra-light smoke tests and drift manifests.** Tiny QA Benchmark++ (Koc, 2025) proposes smoke-test suites for CI workflows. Camuffo et al. (2026) recommend drift-audit manifests for LLM-as-evaluator in strategy research. This proposal extends that to a venue-enforceable standard for any hosted-API dependency.

- **ML observability.** Shankar & Parameswaran (2022) argue for pipeline observability infrastructure. The LDAA is a lightweight, paper-scoped observability artifact.

- **Open models and existing infrastructure.** Open-weight models (Le Scao et al., 2022) largely avoid drift but frontier capabilities often require hosted APIs. OpenAI Evals (OpenAI, 2023) provides general-purpose evaluation; the LDAA is a minimal, paper-specific invariant suite for reproducibility.

**What is missing is a minimal standard that (i) fits conference workflows, (ii) respects closed-model constraints, and (iii) provides a reproducibility "handle" for drift.**

### 4.1. What this paper adds

Prior work provides building blocks: ChangeLists for post-update reporting, contracts for runtime correctness, and auditing methods for drift detection. What is missing is the *standard packaging*: a minimal, paper-attached artifact designed for conference workflows that specifies what evidence is required for reproducible black-box results at a venue like ICML.

## 5. Proposal: ICML Should Require a Lightweight Drift-Audit Artifact for Closed-API-Dependent Results

### 5.1. The artifact: LLM Drift-Audit Artifact (LDAA)

For any submission where the main empirical claims materially depend on a closed or hosted LLM API, ICML should require an **LLM Drift-Audit Artifact (LDAA)** in supplementary material.

A minimal LDAA contains:

1. **Provenance record (machine-readable)**

   - Provider, endpoint, and model identifier string as returned by the API (when available)
   - Date range of API calls
   - Core decoding settings (temperature, top_p, max tokens, any seeds)
   - Prompting scaffolding version (system prompt/template hash if applicable)

- SDK/library versions and any middleware (router, guardrails, tool-calling framework)
- API feature flags under author control (e.g., `tool_choice`, JSON mode, response format)
- Any provider-returned version identifiers or request/trace IDs

2. **A compact audit suite (20–200 cases)**

- Chosen to reflect what the paper actually depends on: schema correctness, refusal patterns, tool-call shapes, etc.
- Each case specifies acceptance criteria as *invariants* (e.g., valid JSON schema, presence/absence of a tool call), not necessarily a single gold answer.

3. **A drift report**

- A small set of metrics aligned to the invariants (pass rate, schema validity, tool-call conformance)
- Optional black-box statistical tests (e.g., inspired by two-sample testing approaches) (Gao et al., 2025)
- Optional low-cost monitoring signals (e.g., logprob-based tests when supported) (Chauvin et al., 2025)

4. **A rerun script**

- One command to re-run the suite and regenerate the report.

## 5.2. Worked example: LDAA for a tool-calling evaluation

Consider a paper that evaluates the tool-calling accuracy of a hosted LLM API on a suite of 500 function-calling tasks, reporting that the model correctly selects and parameterizes the right tool in 87% of cases. Table 1 illustrates what the LDAA provenance record would contain.

The accompanying audit suite might contain 50 cases drawn to cover the paper's key invariants. Each case specifies pass/fail criteria as invariants, not gold outputs. Table 2 illustrates common probe patterns and their evaluation logic.

The drift report records the pass rate for each invariant class at the time of the experiments. A rerun script re-executes the 50-case suite against the live API and compares the new pass rates to the recorded baseline. If the tool-selection pass rate drops below a pre-specified threshold (e.g., from 92% to below 80%), the report flags a potential behavioral drift that may affect the paper's main claims.

*Table 1.* Example LDAA provenance record for a hypothetical tool-calling evaluation paper.

| Field | Example value |
|---|---|
| Provider | OpenAI |
| Endpoint | `/v1/chat/completions` |
| Model ID | `gpt-4-0613` |
| Date range | 2025-08-01 to 2025-08-14 |
| Temperature | 0.0 |
| Seed | 42 |
| Max tokens | 512 |
| SDK version | `openai==1.35.0` |
| System prompt hash | `a3f7c2...` (SHA-256) |
| Feature flags | `tool_choice=auto`, JSON mode off |
| Request/trace ID | `req-abc123...` (if returned) |

*Table 2.* Example probe patterns for an LDAA audit suite.

| Probe class | Pass criterion | Aggregation |
|---|---|---|
| JSON schema validity | Output parses and matches declared schema | Hard fail: any invalid $\Rightarrow$ suite fails |
| Tool selection | Function name matches ground truth | Rate + 95% CI (e.g., $92\% \pm 3\%$) |
| Refusal on safe prompts | Model does not refuse benign queries | Rate + CI; flag if >5% refuse |
| Distribution shift | Function selection distribution stable | Two-sample test ($p < 0.01$) |

## 5.3. Two audit modes: contract vs. behavioral-distribution

Not all API dependencies are equally amenable to binary invariant checks. The LDAA can support two audit modes:

**Contract-mode LDAA.** When the paper depends on well-defined structural properties—JSON schema validity, tool-call argument formats, refusal vs. non-refusal on a fixed set of prompts—the audit suite can use binary pass/fail invariants. This mode is well suited to tool-calling pipelines, structured output generation, and classification tasks where correctness is unambiguous. Contract-mode probes are cheap, deterministic in expectation, and easy to interpret.

**Behavioral-distribution LDAA.** When the paper depends on open-ended semantic quality—summarization fidelity, reasoning coherence, or creative generation—binary invariants are insufficient. In this mode, the audit suite samples representative tasks and evaluates outputs using a scoring function (e.g., an embedding-based similarity, a rubric-based LLM judge, or human ratings). The drift report then compares score distributions rather than pass rates, using sta-

tistical tests (e.g., two-sample $t$-test, Kolmogorov–Smirnov) to flag significant shifts.

Behavioral-distribution mode introduces additional complexity: the scoring function itself may drift if it relies on another hosted API. Authors should either (i) use a fixed open-weight model as the judge and include its checkpoint hash in provenance, or (ii) collect a small set of human reference labels and report agreement with those labels as a stability anchor. This tradeoff between auditability and cost is inherent; the LDAA makes it explicit rather than hiding it.

**Worked example: LLM-as-judge.** Consider a paper that uses GPT-4 to score summaries on a 1–5 rubric and reports that Method A outscores Method B by 0.5 points. The key audit-design insight is that the claim-relevant statistic is the *paired score difference* (A minus B) on a fixed sentinel set, *not* the judge's global calibration: the audit tracks the *margin*, which is preserved under uniform calibration drift of the judge; semantic drift that coincidentally preserves the margin is caught only by the judge-stability mitigations below, not by margin-tracking alone. Concretely, the audit suite is a fixed set of ~50 representative (input, reference) pairs scored by both methods. At submission, the authors record the distribution of paired differences and the pairwise win rate of A over B. The rerun script re-scores the *same* pairs under identical decoding settings—running each probe several times to account for nondeterminism (Section 5.5)—and flags drift if the paired difference or win rate has shifted enough to narrow or erase the reported margin. Because the judge is itself a hosted API, the provenance record and the mitigation options above (fixed open-weight judge, or human-label anchor) apply to the judge as well.

**Choosing the statistic.** The appropriate test depends on the claim type, not on the audit mode in the abstract. Table 3 maps common claim types to a statistic and a drift test. Pass/fail and schema invariants take a pass-rate delta with a confidence interval, distinguishing *hard contracts* (zero-tolerance, e.g., schema validity or safety-critical refusals) from probabilistic pass rates that admit a tolerance band consistent with the nondeterminism budget in Section 5.5; paired rubric scores on the same items call for a paired test on the claim-relevant quantity (as in the worked example), with the $k$ reruns per item aggregated (e.g., mean across reruns) before comparison so that repeated runs are not treated as independent samples; pairwise preferences are captured by a sign test or a flip-rate check; and open-ended claims should distinguish *mean-shift* claims (Welch's $t$-test on the mean) from *distributional-shape* claims (Kolmogorov–Smirnov). The author chooses and *declares* the statistic and decision threshold in the claim–dependency–probe mapping (Section 5.6), so that "behavioral-distribution auditing" re-

*Table 3.* Statistical procedures by claim type for behavioral-distribution audits. The author declares the chosen statistic and threshold in the claim–dependency–probe mapping (Section 5.6).

| Claim / metric type | Statistic | Drift test |
|---|---|---|
| Pass/fail or schema invariant | Pass rate | Pass-rate delta + 95% CI; hard fail only for designated hard contracts |
| Paired rubric scores (same items) | Paired score difference | Paired $t$-test or Wilcoxon on per-item means (aggregate $k$ reruns first) |
| Pairwise preferences | Win rate / flip rate | Sign test or flip-rate threshold |
| Open-ended scores | Mean or full distribution | Welch's $t$-test for mean shift; Kolmogorov–Smirnov for distributional shape |

solves to a specific, reviewable procedure rather than an open-ended intention.

### 5.4. Multi-model and multi-step pipelines

Many LLM-based research pipelines chain multiple API calls—for example, one model generates candidate responses, a second model scores them, and a third extracts structured data. For multi-model pipelines, the LDAA should include a separate provenance record and audit sub-suite for each distinct API dependency, with invariants scoped to the role each model plays in the pipeline. The rerun script should execute each sub-suite independently and report drift per component, since a shift in the scoring model may be invisible to probes designed for the generation model.

This per-component approach avoids the combinatorial explosion of testing every possible interaction, while still providing a diagnostic handle when end-to-end results change.

### 5.5. Why this is feasible and low burden

The audit suite is intentionally small. It can be:

- **Sample-efficient**: API shift estimation via adaptive methods can detect meaningful changes with far fewer queries than exhaustive re-evaluation (Chen et al., 2021).

- **Statistically grounded**: Two-sample testing provides formal guarantees on false-positive and false-negative rates for distributional changes (Gao et al., 2025).

- **Operationally cheap**: Logprob-based monitoring reduces cost substantially by requesting minimal output and using distribution-level signals rather than full generation (Chauvin et al., 2025).

- **Developer-friendly**: Smoke-test suites are designed to run in seconds in CI-like workflows (Koc, 2025). Existing evaluation frameworks such as OpenAI Evals (OpenAI, 2023) provide scaffolding that LDAA tooling can build on.

**Setting drift thresholds.** Authors should choose thresholds based on the sensitivity of their claims. A paper that reports 87% accuracy and claims superiority over a 82% baseline has a 5-point margin; a drift threshold of 3 points on the audit invariants would flag changes that could plausibly close this gap. The LDAA does not prescribe universal thresholds; it requires authors to *declare* their thresholds and the rationale, making the sensitivity assumption explicit and reviewable.

**Handling nondeterminism.** Even with `temperature=0` and a fixed seed, hosted APIs may exhibit run-to-run variation due to batching, load balancing, or backend updates. To distinguish genuine drift from sampling noise: (i) run each probe $k$ times (e.g., $k = 5$) and report the pass rate with a confidence interval; (ii) distinguish *hard failures* (e.g., invalid JSON, missing required fields) from *soft shifts* (e.g., distributional changes in tool selection); (iii) set thresholds that incorporate expected run-to-run variance, not just raw point differences. For example, if a probe passes 92% of the time with a 95% CI of $\pm 4\%$, a threshold of 85% accounts for natural variance while still flagging meaningful degradation.

### 5.6. Aligning audit suites with paper claims

A risk with any artifact requirement is superficial compliance: authors could submit an LDAA that checks trivial properties unrelated to their actual claims. To mitigate this, authors should include a **claim–dependency–probe mapping** that explicitly links each main claim to the API behavior it depends on and the probe class that monitors that behavior. Table 4 illustrates this mapping for the tool-calling example.

This mapping makes the audit suite's coverage reviewable: a reviewer can verify that the probes address the dependencies that matter, not just the dependencies that are easiest to test.

### 5.7. What ICML would and would not enforce

ICML would not need to verify that an audit suite is "perfect." It would only require that (i) authors disclose their dependency as a living API, and (ii) authors provide a minimal, runnable drift handle tied to their claims.

*Table 4.* Example claim–dependency–probe mapping.

| Claim | API dep. | Probe class | Threshold rationale |
|---|---|---|---|
| 87% tool accuracy | Tool selection | Name match | 3pt margin to 82% baseline |
| Valid JSON 100% | Schema | JSON parse | Hard fail: any invalid |

This aligns with ICML's emphasis on scholarship expectations for evidence and attribution, and with the main track's explicit interest in evaluation methodology and replicability.

### 5.8. Integration into the ICML workflow and phased adoption

A natural concern is *when* and *how* authors would produce an LDAA, and what happens if they do not. We propose an integration that mirrors existing ICML processes and an adoption path that escalates expectations over time.

**Submission, review, and camera-ready.** The artifact attaches at points the workflow already has:

- **At submission**: Authors answer a single declaration—do the main claims depend on a hosted LLM API?—analogous to the existing ethics and LLM-usage declarations. If yes, the LDAA (provenance metadata, a claim-aligned probe suite, and a rerun script) is included in supplementary material.

- **During review**: Reviewers check that the claim–dependency–probe mapping (Table 4) covers the paper's actual dependencies. Running the rerun script is optional, not required; the artifact is standardized and checkable, not deeply audited.

- **At camera-ready**: Authors re-run the suite and update the drift report, documenting any API changes during the review period.

This parallels how ICML already handles reproducibility checklists and code submissions, minimizing process disruption.

**Discrete checkpoints, not continuous monitoring.** The LDAA requires only *point-in-time* auditing at discrete checkpoints (submission and camera-ready); it does not ask authors or the venue to monitor an endpoint indefinitely. The rerun script executes once per checkpoint and produces a snapshot report; later reruns are optional diagnostics for authors or readers, not a venue obligation. The monitoring overhead is therefore bounded by two snapshot runs, not a standing service. If an endpoint is retired before camera-ready, that retirement is itself a reportable drift outcome, not

a failure of the artifact. Consistent with Alternative View A and Section 10, the goal is to make drift *diagnosable*, not to guarantee continuous reproducibility of closed-model results.

**Phased, norm-based adoption.** We do not propose to make the LDAA mandatory overnight; adoption can escalate in phases with corresponding incentives:

- **Phase 1 (disclose)**: An optional API-dependency disclosure checkbox and voluntary LDAA submission. The incentive at this stage is self-protective—authors with an LDAA can diagnose drift when results are later questioned, rather than having non-reproduction attributed to methodological weakness.

- **Phase 2 (reward)**: A "Drift-Aware Artifact" badge (Section 9) that recognizes best practice, a positive incentive analogous to existing reproducibility badges.

- **Phase 3 (require)**: A required LDAA for papers whose main claims materially rely on hosted APIs. We do not prescribe a specific penalty mechanism; as with code submission and reproducibility checklists, the appropriate enforcement model—reviewer consideration, area-chair weighting—would emerge from community iteration. In a mature phase, the absence of an LDAA for a materially API-dependent claim would be treated like missing code for a computational claim: missing evidentiary support for a central dependency.

This phased path lets norms and tooling mature together, rather than imposing a hard requirement before the community has converged on what a good LDAA looks like.

# 6. Enabling Infrastructure: Behavioral Versioning and Machine-Readable Changelogs

## 6.1. Why "versioning" must include behavior, not just endpoints

Semantic Versioning requires a clearly defined public API (Preston-Werner, 2013), but for hosted LLMs the "API" includes behavior: structured output reliability, tool-call conventions, and refusal patterns that downstream code depends on (Ajibode et al., 2024; Romel et al., 2025).

## 6.2. Behavioral SemVer (bSemVer) as a provider norm

This paper proposes that providers adopt bSemVer semantics: a **MAJOR** bump for breaks in declared hard behavioral contracts; **MINOR** for added capabilities preserving contracts; **PATCH** for bug fixes preserving contracts. This is not a claim that behavior becomes deterministic—it is

a claim that providers can commit to **contract surfaces** testable probabilistically.

## 6.3. Machine-readable behavioral changelog (MBCL)

Providers already publish changelogs (OpenAI, 2024a; Anthropic, 2024; Google, 2024a). This paper proposes adding a machine-readable layer (Lacan, 2017) with: version identifier, date, contract surfaces changed, deprecation dates, known regressions, and compatibility testing methodology. This would let researchers automatically reason about whether a paper's dependency remained compatible.

## 6.4. Supply chain transparency

ML supply chain standards demonstrate that machine-readable dependency metadata is achievable: CycloneDX supports ML models in bill-of-materials inventories (Ecma International, 2024), FAIR4ML proposes structured metadata (RDA FAIR4ML Working Group, 2024), and recent work maps LLM supply chain dependencies (Wang et al., 2024).

# 7. Predicted Benefits and Falsifiable Evaluation

This position is falsifiable: if adopted, it should improve measurable outcomes. This section states four predictions and describes how each could be tested.

1. **Replication stability.** A higher fraction of closed-API-dependent results should remain qualitatively consistent over time (or at least become diagnosable as drift). *Test*: Compare replication rates for ICML papers with and without LDAAs over a 12-month window after publication. If LDAAs provide no diagnostic value, replication rates and failure-mode classification should be indistinguishable between the two groups.

2. **Time-to-diagnosis.** When results stop reproducing, audits should enable rapid identification of drift versus implementation error. *Test*: For papers that fail to replicate, measure the time from first failure report to root-cause identification. Papers with LDAAs should achieve diagnosis faster, because the rerun script immediately surfaces whether the API's invariant-pass rates have changed.

3. **Transparency improvements.** Providers' disclosures (changelogs, deprecations) become more actionable when aligned to behavioral compatibility (OpenAI, 2024a;b; Anthropic, 2024; Google, 2024a;b). *Test*: Survey researchers who use hosted APIs about the actionability of provider changelogs before and after the introduction of machine-readable behavioral metadata

(if adopted).

4. **Bounded burden.** Artifact cost stays low because audits are small and can be automated. *Test*: Measure the median API cost and author time to produce an LDAA for a sample of submissions. If the median cost exceeds a pre-specified threshold (e.g., $10 or 2 hours of effort), the "low burden" claim is falsified. Existing methods for sample-efficient monitoring (Chen et al., 2021), statistical testing (Gao et al., 2025), log-prob tracking (Chauvin et al., 2025), and smoke-test design (Koc, 2025) all suggest this threshold is achievable, but the claim remains empirical.

## 8. Alternative Views

**A: Reject closed-API papers as irreproducible.** This would exclude frontier-relevant research and bias the field toward what is open rather than what is important. Drift audits make results interpretable and diagnosable—a meaningful improvement even if not full reproducibility.

**B: Versioning is meaningless for stochastic LLMs.** Compatibility can be defined over *contract surfaces* and *probabilistic tolerances*, not exact outputs. Two-sample testing and logprob-based monitoring explicitly treat behavior statistically (Gao et al., 2025; Chauvin et al., 2025).

**C: Safety updates require silent changes.** This proposal does not require indefinite access to old versions—only compatibility-aware disclosure and research provenance. Providers can retire unsafe versions while still supporting scientific interpretability via changelogs and bounded auditing artifacts.

**D: Drift audits burden small labs.** Properly scoped audits are small and automatable (Koc, 2025; Chen et al., 2021). They can protect small labs by reducing wasted time chasing "non-reproducible" results caused by upstream drift.

**E: Providers, not venues, should solve this.** This paper advocates for provider-side improvements in Section 6, but the LDAA works *today* with existing APIs. If venues require drift artifacts, providers gain incentive to supply enabling metadata. The approaches are complementary.

**F: Results expire with the model, so drift audits are unnecessary.** On this view, a closed-model result is only relevant while the underlying model is current: the field moves fast, the model the authors used will have drifted or been retired within a year, and the result's relevance expires with it—so why audit for drift at all? We think this framing is too narrow, in three ways. First, even granting a limited

shelf life, readers still need to know *which* regime they are in. When a result later fails to reproduce, an LDAA lets the community distinguish a genuine weakness in the scientific contribution from a mere upstream API change, protecting both the original authors and later readers from misattributing drift to methodological error (Section 3, Alternative View A). Second, the "expires with the model" intuition assumes results are consulted only while the model is live, but published claims continue to be cited, built upon, and aggregated in meta-analyses long after the endpoint changes; the scientific record outlives the dependency. Third, the cross-disciplinary evidence in Section 2—reproducibility failures documented in ML benchmarking, political science, business research, and science policy—shows the community does not, in practice, treat these results as disposable. The upshot inverts the objection: the *shorter* the shelf life of closed-model results, the more valuable a cheap, point-in-time diagnostic handle becomes, because drift is then the rule rather than the exception.

## 9. Call to Action

### 9.1. For ICML (and similar venues)

1. **Add an LLM/API dependency disclosure section**: Authors state whether core results rely on hosted LLM APIs.

2. **Require LDAA for closed-API-dependent main claims**: A small drift-audit suite plus provenance metadata.

3. **Create an optional "Drift-Aware Artifact" badge**: Reward best practices without blanket exclusion.

### 9.2. For providers

1. **Declare behavioral contract surfaces** (what downstream code can rely on) (Romel et al., 2025).

2. **Adopt behavioral versioning semantics** aligned with the "declare a public API" principle (Preston-Werner, 2013).

3. **Publish machine-readable changelog and deprecation metadata** alongside existing release notes (Lacan, 2017; OpenAI, 2024a;b; Anthropic, 2024; Google, 2024a;b).

### 9.3. For tool builders and the community

1. Build open tooling that generates drift audits from a prompt suite and endpoint configuration (Gao et al., 2025; Chauvin et al., 2025; Koc, 2025).

2. Align drift-audit metadata with broader ML metadata and supply-chain transparency efforts (RDA FAIR4ML

Working Group, 2024; Ecma International, 2024; Wang et al., 2024).

## 10. Scope and Limitations

**Scope and material dependence.**    The LDAA is designed for papers whose *main empirical claims* materially depend on hosted LLM APIs. It is not intended for papers that use LLMs incidentally (e.g., for data preprocessing or text cleanup) or for papers that use open-weight models that can be pinned and redistributed. The boundary between "material dependence" and "incidental use" requires judgment, much like the existing boundary between "main results" and "supplementary experiments."

To guide this judgment, this section offers a decision heuristic. An API dependency is likely **material** if: (i) the LLM's outputs directly determine a reported metric (e.g., the LLM labels data used for training or evaluation); (ii) the paper's main claim would be invalidated if the API's behavior changed (e.g., LLM-as-judge evaluation where the judge scores are the dependent variable); or (iii) the LLM generates synthetic data and the quality of that data is a core claim. An API dependency is likely **incidental** if: the LLM is used for auxiliary tasks that do not affect main results (e.g., qualitative error analysis in an appendix, formatting assistance, or exploratory data inspection).

Gray areas exist: for example, using an LLM to generate a fixed synthetic dataset that is then used for training. If the dataset is released and the paper's claims are about downstream model performance rather than data quality, the LLM dependency may be incidental. If the paper claims the synthetic data itself has desirable properties, the dependency is material. We expect that community norms will refine these boundaries, as they have for related artifact requirements.

**What the LDAA does not solve.**    The LDAA addresses one failure mode—behavioral drift—but does not solve all reproducibility challenges for closed-model research. It does not address data contamination (Balloccu et al., 2024), opaque training data, or the fundamental impossibility of fully reproducing a black-box system. It also cannot detect adversarial model substitution by a provider, a problem that Cai et al. (2025) show is fundamentally hard to solve with software-only methods. The LDAA is a pragmatic minimum: it gives the community a shared, checkable norm for the drift problem specifically.

**Adoption challenges.**    Section 5.8 lays out the workflow and phased adoption path; the residual friction is tooling. Libraries that auto-generate provenance records and stub audit suites from API call logs would substantially lower the barrier for both reviewers and authors. The LDAA is a starting point for community iteration, not a final specification.

**Privacy and safe release of audit suites.**    Audit suites may contain sensitive content: proprietary prompts, personally identifiable information, or adversarial inputs designed to test refusal behavior. Authors should use synthetic or sanitized probes when real prompts cannot be shared publicly. For cases where prompt confidentiality is essential, authors may release hashed prompts with a deterministic rerun mechanism, accepting reduced interpretability as a tradeoff. Venues could also consider an "artifact committee only" pathway for sensitive suites, analogous to how some venues handle proprietary datasets. Special caution is warranted for refusal probes: prompts designed to elicit unsafe behavior should be clearly documented and flagged, and venues may require that such probes be shared only with reviewers rather than publicly released.

**Store-and-replay as a complementary technique.**    The LDAA focuses on *live rerun* of probes against the current API. A complementary approach is to store raw API outputs at publication time, enabling offline re-analysis if the API changes or disappears. Stored outputs support more granular drift diagnosis (comparing old vs. new outputs token-by-token) and provide a fallback if the endpoint is deprecated. However, storing outputs may conflict with provider terms of service, and stored outputs alone cannot detect whether the API *still* behaves as documented. Store-and-replay is a valuable complement to live probes, not a replacement; the LDAA's rerun mechanism remains the primary means of detecting ongoing drift.

## 11. Conclusion

Hosted LLM APIs are becoming core scientific dependencies in ML research, but they drift and deprecate in ways that undermine reproducibility unless we adapt our norms (Chen et al., 2023; Ma et al., 2023; Chen et al., 2021); the resulting failures already span ML benchmarking, political science, business research, and science policy (Barrie et al., 2025; Thomas et al., 2026; Kapoor & Narayanan, 2023).

**ICML should require a lightweight drift-audit artifact for papers whose key results depend on closed LLM APIs, and the community should push toward behavioral versioning and machine-readable behavioral changelogs.** The LDAA proposed here is deliberately minimal—a provenance record, a small invariant-checking probe suite, and a rerun script—and does not solve the full problem of closed-model reproducibility, but it provides a concrete, checkable handle that makes drift diagnosable and results more interpretable. What has been missing is the operational bridge: a minimal, venue-adoptable standard that fits conference workflows and respects closed-model constraints. The LDAA is that bridge.

## Impact Statement

This paper proposes reproducibility infrastructure for ML research that depends on hosted LLM APIs. The proposal aims to improve scientific transparency and replicability. No negative societal consequences specific to this work are foreseen beyond those that are well established when advancing the field of Machine Learning.

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
