# OpenReview forum: "Position: ICML Should Treat Hosted LLM APIs as Versioned Dependencies and Require Drift-Audit Artifacts"
_ICML.cc/2026/Position_Paper_Track — ICML 2026 Position Paper Track regular_

### Official Review · Reviewer_RxP4 · 2026-03-12

**Significance:** 3
**Argument Clarity:** 3
**Rating:** 4
**Confidence:** 3

**Questions:**

See the above weakness discussion.

**Alternative Views Section:**

Yes

**Compliance With Llm Reviewing Policy A Conservative:**

Affirmed.

**Discussion Potential:**

3

**Final Justification:**

I read the authors' rebuttal and also other comments. I think the authors have perfectly addressed my main concerns. So I will keep my positive score like other reviewers.

**Paper Summary:**

In this paper, the authors argue that conferences like ICML should require a new reproducibility standard—the LLM Drift-Audit Artifact (LDAA) —for research that depends on hosted LLM APIs since the LLM APIs can become deprecated due to silent updates and behavioral drift, which can hurt the reproducibility of the ML papers relying on these APIs. The proposal is designed to be low-cost, statistically grounded, and compatible with closed models, giving the community a practical, checkable tool to make drift diagnosable and results more interpretable, while also calling on providers to adopt behavioral versioning and machine-readable changelogs.

**Position:**

Yes

**Position In Title:**

Yes

**Related Work:**

3

**Strengths And Weaknesses:**

Strengths:
+ The authors reveal an important reproducibility issue due to the widely use of LLM APIs in the state-of-the-art ML papers, which can become deprecated silently on the provider side.
+ The authors extensively review related works, which can benefit the claims in the paper.
+ The authors not only give suggestions of how to resolve the deprecated API issues, but also analyze potential outcome if the suggested actions are taken, which is very comprehensive.

Weakness:
+ One biggest concern of mine is that the proposed action may add extra burden to when people submit papers. Although the authors mentioned that the extra artifacts provided only required minimal efforts, they did not give suggestions of when and how people should provide such artifacts in the paper submission and reviewing process.
+ Related to the above weakness point, the authors did not discuss how to motivate people to provide such additional artifacts or how to penalize those who fail to provide them. More discussions of incentives would be helpful.
+ In addition, even if people can provide such artifacts, it seems that monitoring whether the reproducibility can be guaranteed is inevitable. The overhead of performing such monitoring is not extensively studied in the paper

**Support:**

3

---

> ### Author Rebuttal · Authors · 2026-03-31
>
> We thank Reviewer RxP4 for the positive assessment and the practical questions about adoption.
>
> **On submission workflow and burden (W1, W3):**
>
> An important scope point first: the LDAA applies only to papers whose *main empirical claims materially depend* on hosted LLM APIs. Section 10 provides a concrete decision heuristic for distinguishing material from incidental API use; incidental uses (e.g., preprocessing, formatting) do not require an LDAA. This substantially narrows the set of papers affected.
>
> Within that scope, we propose a concrete integration that mirrors existing ICML processes:
>
> - *At submission*: Authors declare whether main claims depend on hosted APIs (a checkbox, analogous to the existing ethics and LLM policy declarations). If yes, the LDAA (provenance metadata, a small claim-aligned probe suite, and a rerun script) is included in supplementary materials.
> - *During review*: Reviewers verify that the claim–dependency–probe mapping (Table 3) covers the paper's actual dependencies. Reviewers can optionally run the rerun script but are not required to do so; the artifact is standardized and checkable, not deeply audited.
> - *At camera-ready*: Authors re-run the audit suite and update the drift report, documenting any API changes during the review period.
>
> An important framing point: as we state in Alternative View A (Section 8) and Section 10, the LDAA is a pragmatic diagnostic handle for one failure mode: it aims to make drift *diagnosable*, not to guarantee full reproducibility of closed-model results. Accordingly, it requires only *point-in-time* auditing at discrete checkpoints (submission and camera-ready); later reruns are optional diagnostics for authors or readers, not a venue obligation. The rerun script executes once per checkpoint and produces a snapshot report. If the endpoint is retired before camera-ready, that retirement is itself a reportable drift outcome, not a failure of the artifact. Prediction 4 (Section 7) frames bounded burden as a falsifiable claim: if the median cost exceeds $10 or 2 hours of author effort, the "low burden" claim is falsified. Section 5.5 presents existing sample-efficient methods that suggest this threshold is achievable, but the claim remains empirical.
>
> This parallels how ICML already handles reproducibility checklists and code submissions, minimizing process disruption.
>
> **On incentives and penalties (W2):**
>
> The reviewer rightly asks about both motivation and penalties. We envision phased, norm-based adoption with escalating expectations:
>
> - *Phase 1*: Optional API-dependency disclosure checkbox and voluntary LDAA submission. At this stage the incentive is primarily self-protective: authors with LDAAs can diagnose drift when results are questioned, rather than having non-reproduction attributed to methodological weakness.
> - *Phase 2*: A "Drift-Aware Artifact" badge (Section 9, Call to Action) recognizing best practices, a positive incentive that rewards transparency, analogous to existing reproducibility badges.
> - *Phase 3*: Required LDAA for papers whose main claims materially rely on hosted APIs, as proposed in Section 9. The paper does not prescribe a specific penalty mechanism, and we believe the appropriate enforcement model would emerge from community iteration, much as it did for code submission and reproducibility checklists. Existing precedents (NeurIPS code submission, ICML reproducibility checklist) suggest that reviewer consideration and area-chair weighting are natural mechanisms. In a mature phase, absence of an LDAA for materially API-dependent claims would be treated as missing evidentiary support for a central dependency, comparable to missing code for a computational claim.
>
> We will add this phased adoption discussion to the final version.

---

> > ### Author Rebuttal · Reviewer_RxP4 · 2026-04-03
> >
> > Thanks for the authors' rebuttal. My concerns have been fully addressed. I will keep my positive score.

---

### Official Review · Reviewer_bGdr · 2026-03-13

**Significance:** 3
**Argument Clarity:** 3
**Rating:** 4
**Confidence:** 4

**Questions:**

Q1. The current approach seems better fit structured or constrained tasks. How do the authors envision defining and conducting drift audits for less standardized settings such as open-ended generation, reasoning tasks, or LLM-as-judge evaluations?

Q2. The paper mentions behavioral-distribution auditing, but its operationalization remains unclear. Could the authors provide more concrete details on the statistical metrics, decision thresholds, and practical procedures involved?

**Alternative Views Section:**

Yes

**Compliance With Llm Reviewing Policy A Conservative:**

Affirmed.

**Discussion Potential:**

3

**Paper Summary:**

This position paper argues that ICML should require a lightweight LLM Drift-Audit Artifact (LDAA) for papers whose key empirical claims rely on hosted LLM APIs. The authors argue that hosted APIs behave as evolving external dependencies whose behavior may change over time, which threatens reproducibility even when prompts and evaluation pipelines remain fixed. To address this issue, the paper proposes LDAA, a minimal artifact containing provenance metadata, a probe suite aligned with the paper’s claims, a drift report, and a rerun script. The paper also discusses two auditing modes—contract-based checks and behavioral-distribution monitoring—and suggests that conference policies could encourage more transparent API versioning and change reporting. The paper outlines a minimal SBOM schema, illustrates how SBOM metadata could support automated governance controls, and discusses adoption strategies and research directions for integrating such artifacts into ML system releases.

**Position:**

Yes

**Position In Title:**

Yes

**Related Work:**

3

**Strengths And Weaknesses:**

Strengths:
- Clear Problem Motivation. The paper addresses an important reproducibility issue: hosted LLM APIs can change over time, while current publication norms often treat them as fixed dependencies.
- Concrete Suggestions. Rather than only raising a concern, the paper proposes a specific artifact design, the LDAA, including provenance metadata, probe suites, drift reports, and rerun scripts.
- Clear Organization. The paper is generally well structured and easy to follow, with a clear progression from problem motivation to artifact design and policy recommendations.

Weaknesses:
- Limited technical novelty. The proposed artifact mainly combines practices that already exist in current ML workflows (e.g., provenance tracking, probe-based regression tests, rerun scripts, and API change monitoring). While organizing these elements into a conference-level artifact requirement is useful, the contribution appears primarily organizational rather than introducing a new technical method.

- Limited applicability to open-ended tasks. The proposal is most straightforward for structured outputs (e.g., tool calling or schema-constrained tasks). For open-ended generation settings such as summarization, reasoning, or LLM-as-judge evaluations, the suggested behavioral-distribution auditing is less clearly defined and may be difficult to standardize in practice.

**Support:**

3

---

> ### Author Rebuttal · Authors · 2026-03-31
>
> We thank Reviewer bGdr for the detailed and constructive review.
>
> **On technical novelty (W1):**
>
> The paper's contribution is a complete artifact specification and policy proposal that did not previously exist: a scoped artifact structure (the LDAA), a material-dependence heuristic for determining when it applies, a claim-dependency-probe mapping for reviewable coverage, two distinct audit modes (contract and behavioral-distribution), and falsifiable predictions for evaluating the proposal's impact. Section 4 (Related Work) and Section 4.1 (What This Paper Adds) discuss how prior work on change disclosure, contracts, and auditing informs this design, but none of that work proposes a venue-adoptable artifact standard. The ICML position paper track invites papers that argue for what the community *should* do, and we believe a concrete, venue-adoptable standard is the appropriate contribution for this track.
>
> **On open-ended tasks (W2/Q1):**
>
> Our intent was for Sections 5.3 and 5.5 to cover this case, but we agree the paper would benefit from a more concrete operational template. We will add one in the final version. Here is the kind of example we plan to include:
>
> **Worked example (LLM-as-judge):** A paper uses GPT-4 to score summaries on a 1–5 rubric and reports that Method A outscores Method B by 0.5 points. The LDAA audit suite consists of ~50 representative (input, reference) pairs scored by both methods. The claim-relevant statistic is the paired score difference (A minus B) on this sentinel set; the audit tracks *that* quantity, not just the judge's global calibration. At submission, the authors record the distribution of paired differences and the pairwise win rate of A over B. The rerun script re-scores the same pairs under identical decoding settings (running each probe multiple times to account for nondeterminism, as discussed in Section 5.5) and checks whether the paired difference or win rate has shifted enough to narrow or erase the reported margin.
>
> Section 5.3 also flags that the scoring function itself may drift if it relies on another hosted API, and recommends two mitigations: using a fixed open-weight model as the judge (with checkpoint hash in provenance), or anchoring against human reference labels.
>
> Open-ended tasks are inherently harder to audit than structured tasks, and the LDAA reflects that by making the difficulty explicit and reviewable.
>
> **On statistical details (Q2):**
>
> We agree this needs to be more concrete. The paper specifies two-sample tests and author-declared thresholds (Section 5.3) with repeated runs to bound nondeterminism (Section 5.5), but does not map specific test choices to claim types. In the final version, we plan to add guidance along these lines: for pass/fail or schema invariants, a pass-rate delta with confidence interval is sufficient; for paired rubric scores on the same items (as in the worked example above), a paired test on the claim-relevant quantity is appropriate; for pairwise preferences, a sign test or flip-rate check captures whether item-level rankings have shifted. The exact statistic should be chosen by the author and declared in the claim-dependency-probe mapping (Section 5.6).

---

> > ### Author Rebuttal · Reviewer_bGdr · 2026-04-03
> >
> > I thank the authors for their response.

---

### Official Review · Reviewer_HAXk · 2026-03-13

**Significance:** 3
**Argument Clarity:** 4
**Rating:** 5
**Confidence:** 4

**Questions:**

N/A

**Alternative Views Section:**

Yes

**Compliance With Llm Reviewing Policy A Conservative:**

Affirmed.

**Discussion Potential:**

3

**Final Justification:**

As already mentioned, the paper presents a simple yet sensible position that it defends in a coherent and easy-to-digest manner. The authors engaged with my comment in their rebuttal, which I'm grateful for, and I can say that I see their point of view.

Hence, I maintain my rating in support of accepting this paper to the conference.

**Paper Summary:**

The authors focus on the problem of drift of closed-model LLM APIs and propose that conferences require a drift-audit artifact to be submitted with every paper featuring results that depend on LLM APIs. The paper presents an overview of the state of the field, highlighting the negative impact of model drift on paper reproducibility, propose an additional artifact to be required as part of paper submissions. They also present an array of alternative approaches and argue why their proposed approach strikes a balance between practicality and low effort.

**Position:**

Yes

**Position In Title:**

Yes

**Related Work:**

4

**Strengths And Weaknesses:**

**Strengths:**

**(S1)** The stated position is relatively simple, sensible, and thus easily defensible.

**(S2)** The paper is well-written, easy to follow, and states clear objectives, scope, as well as falsifiable hypotheses that can be used to judge the merit of their claims.

**Weaknesses:**

**(W1)** The problem of paper reproducibility is taken as a self-evident imperative. However, one can claim that when studying closed-model LLMs, this comes with the territory. If we think of the speed at which the field is moving, a paper will conduct experiments today, present at a conference 6 months from now, and 12 months from now, the model the paper authors used will have drifted away, and with it the relevance of the results. To put it bluntly, a result in a paper is relevant as long as the model it was based on is relevant. I realize that I am presenting a counterargument here, which is why this is not a very strong weakness of the paper. Nevertheless, perhaps it is something the authors might try to address in a final version of their paper.

**Support:**

3

---

> ### Author Rebuttal · Authors · 2026-03-31
>
> We thank Reviewer HAXk for the positive assessment and for raising this thoughtful counter-argument.
>
> **On W1 ("a result is relevant as long as the model it was based on is relevant"):**
>
> We agree that hosted-model results may naturally have a shorter shelf life. But that is precisely why dependency-aware reporting matters *more*, not less. As the paper argues in Section 3 (existing norms leave a gap at time t+delta) and Alternative View A (Section 8), the LDAA accepts that dependencies change, and instead makes the source and character of drift visible so that later readers can interpret claims correctly. When a result stops reproducing, the community needs to distinguish whether the failure reflects a weakness in the scientific contribution or simply an upstream API change. That ambiguity harms both the original authors and the community.
>
> We also note that Section 2.6 documents that hosted API drift is a cross-disciplinary reproducibility problem, with failures documented in political science (Barrie et al., 2025), business research (Thomas et al., 2026), and research policy (Kapoor & Narayanan, 2023). This suggests the "results expire with the model" framing may be too narrow.
>
> We will sharpen this point in the final version.

---

> > ### Author Rebuttal · Reviewer_HAXk · 2026-04-01
> >
> > I thank the authors for their response. Having read through the rebuttal and other reviews, I will maintain my current rating in support of having this paper accepted.

---

### Decision · Program_Chairs · 2026-04-30

**Decision:**

Accept (regular)

**Comment:**

All reviewers agreed that this paper addresses a critical and timely reproducibility issue in modern ML research—the silent deprecation and behavioral drift of closed-model LLM APIs. The proposed solution, a lightweight LLM Drift-Audit Artifact, is practical, well-structured, and appropriately scoped as a policy recommendation for the community. During the rebuttal, the authors effectively resolved reviewer concerns regarding the auditing of open-ended tasks and the potential submission burden by outlining a phased adoption roadmap and concrete statistical guidelines. Given its relevance to evaluation methodologies, unanimous reviewer support, and potential to spark necessary community discussion, an accept is recommended.